# Occurrence and Concentration of Chemical Additives in Consumer Products in Korea

**DOI:** 10.3390/ijerph16245075

**Published:** 2019-12-12

**Authors:** Syed Wasim Sardar, Younghun Choi, Naree Park, Junho Jeon

**Affiliations:** 1Graduate School of FEED of Eco-Friendly, Offshore Structure, Changwon National University, Changwon, Gyeongsangnamdo 51140, Korea; syedwasim336@gmail.com (S.W.S.); dudgnsdlsl@naver.com (Y.C.); cooco526@nate.com (N.P.); 2School of Civil, Environmental and Chemical Engineering, Changwon National University, Changwon, Gyeongsangnamdo 51140, Korea

**Keywords:** consumer products, isothiazolinones, phthalates, additives, suspect and non-target screening, exposure assessment, LC-HRMS

## Abstract

As the variety of chemicals used in consumer products (CPs) has increased, concerns about human health risk have grown accordingly. Even though restrictive guidelines and regulations have taken place to minimize the risks, human exposure to these chemicals and their eco-compatibility has remained a matter of greater scientific concern over the years. A major challenge in understanding the reality of the exposure is the lack of available information on the increasing number of ingredients and additives in the products. Even when ingredients of CPs formulations are identified on the product containers, the concentrations of the chemicals are rarely known to consumers. In the present study, an integrated target/suspect/non-target screening procedure using liquid chromatography-high resolution mass spectrometry (LC-HRMS) with stepwise identification workflow was used for the identification of known, suspect, and unknown chemicals in CPs including cosmetics, personal care products, and washing agents. The target screening was applied to identify and quantify isothiazolinones and phthalates. Among analyzed CPs, isothiazolinones and phthalates were found in 47% and in 24% of the samples, respectively. The highest concentrations were 518 mg/kg for benzisothiazolone, 7.1 mg/kg for methylisothiazolinone, 2.0 mg/kg for diethyl phthalate, and 21 mg/kg for dimethyl phthalate. Suspect and non-target analyses yielded six tentatively identified chemicals across the products including benzophenone, ricinine, iodocarb (IPBC), galaxolidone, triethanolamine, and 2-(2H-Benzotriazol-2-yl)-4, 6-bis (1-methyl-1-phenylethyl) phenol. Our results revealed that selected CPs consistently contain chemicals from multiple classes. Excessive use of these chemicals in daily life can increase the risk for human health and the environment.

## 1. Introduction

The development of the chemical industry in the past century has introduced a vast amount of chemicals to the world. Presently, there are approximately 100,000 chemicals being used globally and over 500 new chemicals are produced annually [1]. Human exposure to these chemicals occurs in many environments and along several pathways. For the majority of chemicals, the main exposure pathway takes place in indoor environments through consumer products (CPs) [2]. CPs are widely used in daily life for personal hygiene, home care, and disinfection which contain an excessive amount of chemicals [3]. These chemicals are used as active ingredients, preservatives, solvent, or additives [1]. For instance, isothiazolinone type biocides are a group of preservatives used for the control of microorganisms in a variety of products such as cleaning agents, fabric softeners, shampoos, toiletries, and other hair and skincare products [4]. Some by-products, impurities, and unintentionally added substances (often from plastic containers) are also included [5]. Each CPs can come into contact with humans either through direct exposures or emissions to the environment media [6]. Human skin is an interface for chemicals to enter the human body through percutaneous absorption [5]. Exposure of humans to these chemicals can cause adverse health outcomes, including reproductive inhibition, endocrine disruption, cancer, immune dysfunction, allergies, skin rashes, eye irritation, respiratory problems, and other chronic diseases [4,7]. Recently in South Korea, 52 deaths and 122 injuries occurred due to the inhalation of aerosolized water containing disinfectants from a humidifier that led to serious lung injuries [3]. Moreover, around 23 million people in the United States of America are currently affected by asthma which has been suspected as a consequence of chemical exposure via consumption of CPs [8]. According to the National Report on Human Exposure to Chemicals, most of America’s population in every age group have detectable levels of phthalate metabolites, bisphenol A, triclosan, and other common endocrine-disrupting chemicals in their urine [9]. Furthermore, after using these CPs, chemicals are released directly or indirectly to the environment through wastewater treatment plants resulting threats on the ecosystem, mainly chemicals that are persistent and cause ecotoxicity [10,11]. Previous reports indicated that various chemicals used in CPs have been detected in surface waters and sediments with a concentration level of ng/L and ng/g (dry weight), and in some highly contaminated rivers where the concentration have reached up to µg/L and mg/kg (dry weight). Hence it can cause cytotoxicity, neurotoxicity, enzymatic, and genetic toxicity to certain aquatic organisms [12]. Alkylphenols, phthalates, flame retardants, parabens, and polychlorinated biphenyls are also detected in air and house dust with 13–28 and 6–42 compounds, respectively [13].

A major challenge in the management of human and environmental health risk is the lack of available information on the occurrence and concentration of chemicals in CPs [11]. Even when ingredients of consumer product formulations are identified on the product container, the concentrations of these chemicals are rarely known to consumers, public health officials, and scientists [2]. Recently, advanced LC-MS technologies have been used for the analysis of known and unknown compounds in complex matrices [14]. High-resolution mass spectrometers (HRMS), such as Orbitrap, are an available analytical tool for tentative identification of unknown compounds based on accurate mass measurements [15]. It has also been proved to be a promising technique for simultaneous identification and quantification of chemicals at low concentrations in complex samples [16]. Moreover, the chemical screening can be done more reliably even without reference standards by the approach called suspect screening (exact mass as a priori information) and non-target screening (no previous data of the compound is available) [17]. These novel analytical methods enable broad investigation into potentially thousands of chemicals in CPs sample [2].

In the present study, an integrated screening procedure based on liquid chromatography-high resolution mass spectrometry (LC-HRMS) was applied for the identification of hazardous chemicals in selected CPs including personal care products, cosmetics, and washing agents, starting from a quantitative target screening approach to get insight of the occurrence of known chemicals such as isothiazolinones (i.e., Methylisothiazolinone (MI), methylchloroisothiazolinone (CMI), and benzisothiazolinone (BIT)) and phthalates (i.e., diethyl phthalate (DEP) and dimethyl phthalate (DMP)). Furthermore, suspect screening workflow with a stepwise identification scheme was applied for suspected chemicals whereas a non-target screening procedure involving statistical analysis of the data was used for tentative identification of some unexpected chemicals.

## 2. Materials and Methods

### 2.1. Standards and Reagents

All reference standards for target compounds were of high purity grade (>99%). Methylisothiazolinone (MI), benzisothiazolinone (BIT), dimethyl phthalate (DMP), diethyl phthalate (DEP), and their internal standards (dimethyl phthalate-3,4,5,6-d4 and diethyl phthalate-3,4,5,6-d4) were obtained from Sigma-Aldrich, methylchloroisothiazolinone (CMI), and its internal standard benzoisothiazol-3-one-13C6 was supplied by Toronto Research Chemicals. Other solvents, including methyl tert-butyl ether (MTBE), were purchased from Duksan Pure Chemicals, South Korea. Dichloromethane (DCM), methanol, and ethyl acetate were purchased from Fisher Scientific Korea Ltd. HPLC grade water was supplied by Avantor Performance Materials Korea. Individual stock solutions of all target compounds and internal standards were prepared in ethanol and stored at −20 °C before use.

### 2.2. Sample Collection

A total of 85 popular and frequently used CP samples were purchased from major supermarkets in Changwon city Gyeongsangnam-do province South Korea in February 2018 to September 2018. The selection of CPs was based on high consumption rates by checking the ranks of all products through online shopping sites. The samples were classified according to their usage and grouped into 8 categories: shampoos (*n* = 10), body wash (*n* = 10), face cleanser (*n* = 10), dishwasher detergents (*n* = 15), laundry detergents (*n* = 15), fabric softeners (*n* = 15), lipsticks (*n* = 5), and hair dyes (*n* = 5). All samples were stored at 4 °C before use.

### 2.3. Sample Preparation

All samples were extracted using the methodology previously described by Guo and Kannan, 2013 [18], with a minor modification. In brief, a 0.2 g sample was weighted and spiked with 100 ng of internal standards and then extracted with 5 mL of MTBE by shaking in an orbital shaker for 30 min. The mixture was centrifuged at 3500 rpm for 20 min, and the supernatant was collected in another glass tube. The extraction procedure was repeated, and the 10 mL combined extracts were dried under a gentle stream of nitrogen and was reduced to 1 mL and filtered with a 0.45 µm membrane filter and transferred into a glass vial for instrumental analysis.

### 2.4. Liquid Chromatography—High-Resolution Mass Spectrometry 

Quantitative determination of target substances was carried out by using the Ultimate 3000 UPLC system (Thermo Fisher Scientific, San Jose, CA, USA) coupled to QExactive plus Orbitrap (Thermo Fischer Scientific Corporation). Chromatographic separation was performed by reversed-phase X Bridge C18 column (2.1 mm × 50 mm, particle size 3.5 µm, Waters, Milford, MA, USA). The instrumental parameters are shown in Appendix A. The sample injection volume was 10 μL. Water (solvent A) and methanol (solvent B) both acidified with 0.1% formic acid was used as mobile phases. The gradient elution started at 5% of (B) increased to 75% at 10 min then the content of B component was further increased to 95% at 15 min and this condition was held for 5 min, following this mobile phase composition was set-back to initial conditions and maintained for 10 min to equilibrate the column. Mass spectrometry analysis was done by a high-resolution mass spectrometer (QExactive plus Orbitrap), with heated electrospray ionization (HESI) operating in positive mode. Each sample was analyzed in a positive mode, with the following parameters. Ion source: HESI; spray voltage: 3800 V/3000; sheath gas flow: 45 L/min; capillary temperature: 320 °C; heater temp: 50 °C and auxiliary gas flow rate: 10 AU.

### 2.5. Quality Assurance and Quality Control

Quality assurance and control measures were taken to ensure the accuracy of the analytical method. The MS/MS accuracy was calibrated by using the calibration mixtures covering the m/z values of target compounds. To eliminate the leftover of the target analytes between samples, the LC syringe was double washed before and after the injection with MeOH. For every set of 20 samples, procedural blank and spiked blank were analyzed to estimate the background contamination and measurement precision, respectively. No chromatographic peak of the target analytes was found above LOD in the procedural blanks. A ten-point calibration curve over the range of 0.01 to 100 ng/mL was used for linearity evaluation and quantification. The regression coefficient (r^2^) value of the calibration curve was >0.99. The limit of detection (LODs) and limit of quantitation (LOQs) are given in Appendix A. The LODs and LOQs were calculated based on the signal/noise ratio for a standard of known concentration. Concentrations below the LOQ were selected as zero for data analysis.

### 2.6. Data Processing

#### 2.6.1. Target Screening

In target screening compound information and standards are already available, and can be included within a defined MS method and be checked in the routine analysis [19]. The software-package Xcalibur 4.0 (Thermo Fisher, USA) was used for target screening by searching the exact masses of target compounds at known retention times and integrating the peak areas. Accurate mass ion chromatograms and peak lists were created from full scan spectra by using a software, TraceFinder 4.0 (Thermo Fisher, USA). For confirmation of positive findings, MS/MS fragments were used, and compounds were quantified by calibration curve using internal standard calibration with the help of the software.

#### 2.6.2. Suspect Screening

For the identification of suspect compounds, specific information of compound, e.g., structure, molecular formula, isotopic pattern, and mass spectra were used [20]. Environmental Food Safety (EFS) (Thermo Fisher Scientific, San Jose, CA, USA), database was used as the primary source for the suspect list. The database consists of 1729 compounds from a wide range of classes including, artificial flavors, biocides, UV filter, preservatives, and other industrial chemicals. Exact m/z values of the selected compounds were included in the suspect list for triggering data-dependent MS/MS fragmentation between full high-resolution mass scanning of Orbitrap. Spectral information for the suspects collected from EFS was reviewed and compared with the measured data for the tentative identification of compounds.

The raw data obtained from the LC-HRMS analysis were processed by using software TraceFinder 4.0 (Thermo Fisher Scientific, San Jose, CA, USA). Suspected peaks were isolated with criteria such as: mass tolerance of 5ppm; S/N ratio threshold was 3; a minimum peak intensity of 5xE5; isotopic pattern matches more than 70%; manual inspection for peak shape. Other threshold values (e.g., peak area, fragment matching and isotopic fit score,) were also applied for peak identification after subtraction of blank. Further, the MS/MS spectra of the suspects were compared with the library spectrum provided from MassBank and MzCloud. The suspect screening procedure was applied to all compounds. If all the criteria were satisfied, the peaks were taken as suspect compounds. 

#### 2.6.3. Non-Target Screening

The non-target screening was carried out by using data analysis software (Compound Discoverer 2.0, ThermoFisher). For non-target substances, Compound Discoverer 2.0 applies a peak picking algorithm as being used in TraceFinder 4.0. After peak detection, plausible molecular formulas were assigned to the selected peaks with a combination of the following elements: H, C, N, O, S, P, Cl, K, Na, F, and Br. If the MS/MS data of the detected peaks were obtained, fragmentation data for compounds with the assigned formula were examined in the software connected to MS/MS library (i.e., MzCloud). After matching the measurements, the software suggested the best fit compound for the non-target peaks. For further verification for non-target screening, peak information (e.g., peak intensity, peak shape, and retention time) for proposed compounds from both automated software were twice checked with a software called Xcalibur 4.0 (Thermo Fisher Scientific, San Jose, CA, USA). Fragmentation patterns and spectra data were also reviewed again with another library, MassBank and an in-silico fragmentation database, MetFrag. In the first step, irrelevant peaks were excluded, and peaks that were not present in the blank samples or target and suspect lists were selected based on the presence of distinctive isotopic patterns and intensity. The most reasonable molecular formula was determined for the selected peaks. In a second step, library (MassBank and/or MetFrag) searches were performed for matching the proposed components with existing entries in the library.

## 3. Results and Discussion

### 3.1. Target Screening

In total, 85 CP samples were analyzed to quantify 5 target compounds by LC-HRMS Orbitrap as described before. The concentrations and detection frequency of target compounds are shown in Table 1 and the individual concentrations are shown in Appendix A.

#### 3.1.1. Isothiazolinones

Isothiazolinones were detected in 40 (concentration above the LOQ in only 30 products) out of 85 products. CMI, recently banned in household products such as cleaning materials and wet tissues under K-REACH [21], was not detected in any samples. The detection frequency of BIT was 30%, showing the highest rate among targeted isothiazolinones while MI, also forbidden along with CMI was found in 15% of the samples. Across analyzed CPs, BIT was detected in all laundry detergent samples (100%) and frequently found in fabric softener (67%), while MI was detected in body washes (100%) and fabric softener (53%). Furthermore, laundry detergents and fabric softener contained a high concentration of BIT and MI with a concentration in the range of 0.2 to 518 mg/kg and 1.2 to 7.1 mg/kg, respectively. Shampoos contained relatively low concentrations of BIT with a range of <LOQ to 0.165 mg/kg. Any presence of targeted isothiazolinones was not evident in face cleansers, dishwashers, lipsticks, and hair dyes. From the previous study in Switzerland, the reported levels of BIT in laundry detergent were comparable or lower (mean concentration, 82.2 mg/kg) than the concentration measured in the present study, whereas the level of MI (mean concentration, 41.1 mg/kg) was higher [22]. According to the EU Regulation, BIT is not allowed in leave-on and rinse-off cosmetics, while MI and CMI can still be used with maximum permissible concentrations of 100 mg/kg and 15 mg/kg, respectively. Following the aforementioned regulation, the observed maximum concentrations of isothiazolinones in cosmetics and personal care products were in the lower limit and would not be expected to induce skin sensitization. However, there is no maximum concentration defined or permissible concentration set for any of isothiazolinones in household detergents [23], which could have triggered the relatively high concentration of BIT in washing agents (<518 mg/kg). Even though BIT is considered a milder sensitizer than MI and CMI [24], it is concerning that washing agents constitute the second most significant allergen source of isothiazolinones [23]. For exposure assessment not only the concentrations in the products are important but various other factors also need to be considered, such as the type of contact between consumer and product and the frequency of use.

#### 3.1.2. Phthalates

Among the eight categories of CPs analyzed, phthalates were detected in 16 (concentration above the LOQ in only 10 products) out of 85 products. These were frequently found in lipstick and face cleansers, and rarely detected in dishwasher detergents, fabric softeners, and shampoos, and not detected in body washes, hair dyes, and laundry detergents. DMP was found in 16% of the samples whereas DEP was seldom found at percentages <10%. The concentration ranges of DMP and DEP in all samples were from 0.26 to 21.2 mg/kg and from 0.4 to 2.0 mg/kg respectively. The maximum concentration of phthalates was found in lipsticks, 21.2 mg/kg for DMP and 1.1 mg/kg for DEP followed by face cleanser with 8.0 mg/kg of DMP.

The relatively high concentrations of phthalates found in lipsticks and face cleansers are probably due to intentional additions into the product as phthalates have often been used as a softener in cosmetics and personal care products [25]. However, the low detection rate in other CPs suggests that the phthalates might be migrated from plastic packaging materials, as phthalates are the major plasticizer used in polyvinyl chloride (PVC) plastics. A recent report indicated that phthalate levels in urine were significantly reduced by avoiding the consumption of foods that were packaged in plastic materials, which further shows that phthalates can migrate from plastic packaging materials [18]. This result confirms the findings of several studies in which DEP was the most frequently detected phthalate in personal care products and cosmetic [26]. The use of CPs has been suggested as one of the major exposure pathways of phthalates. Previous studies have also shown that phthalates have the highest association with CPs, mostly with the use of perfumes. In Canada, the highest DEP concentration measured in fragrances was 25,500 mg/kg, whereas the highest DEP concentration of perfume was 3960 mg/kg. DMP and DEP concentrations in lipsticks measured in the present study were much less than the Canadian study. However, considering the application features of lipsticks, direct oral ingestion of phthalates can be significant, which is comparable or even more risky than the exposure route of perfumes. In another report showing the exposure consequence, it has been estimated that the total daily intake of DMP for Chinese population is 5.1 mg/kg/day and DEP is 44.4 mg/kg/day through personal care products [18].

### 3.2. Suspect and Non-Target Screening

The LC-HRMS based screening tentatively identified two suspect and four non-targeted compounds. Among those identified with evaluations and interpretations for chromatographic peak, isotopic pattern, and MS/MS fragment (Figure 1 and Figure 2), one suspect (benzophenone) and two non-targets (triethanolamine and galaxolidone) were confirmed with authentic reference standards. According to identification confidence levels suggested by Schymanski et al. 2014 [27], those confirmed acquired level 1. Even though the three remains (ricinine, idocarb, and 2-(2H-Benzotriazol-2-yl)-4,6-bis(1-methyl-1-phenylethyl) phenol) were not fully confirmed due to the lack of standards, but placed in level 2, indicating that the overall identification and evaluation procedures of the suspect and non-target screening are reliable.

Two identified substances via suspect screening, benzophenone and ricinine are often used as additives in personal care products. At the first stage of the suspect screening, peak information for the identifications are described in Figure 1. The peak lists with all evident suspects were exported from the processing software. Then, the appearance and intensity of peaks and similarity of the acquired MS/MS spectra with available libraries (MassBank and MzCloud) were manually evaluated. All compounds detected in blanks were subtracted to reduce false positive. After fulfilling the above criteria, only two substances (benzophenone and ricinine) remained as suspect compounds. Benzophenone was detected in 33% of the samples and was mostly found in dishwashing detergents, fabric softener, face cleaner, body washes, and lipsticks. It is mainly used as sunscreen agent/ultraviolet filters and stabilizer in cosmetics, but also used as a stabilizing agent for plastics and rubber to prevent polymer degradation [32]. Thus, there has been concern about the leaching of benzophenone from the packaging material to edible contents, leading to humans ingestion [33]. Widespread human exposure to benzophenone occurs mainly through the skin and is excreted mostly via urine [34]. Benzophenone type UV filters showed both estrogenic, anti-androgenic, and potential genotoxic effects in various organisms [35]. Meanwhile, ricinine was detected in three lipsticks. The plant originated alkaloid is used as stabilizers, skin-conditioning agents, emulsion, and surfactants in cosmetics [36]. Due to its relatively high toxicity (LD50 value for mice is 2–3 μg/kg), the Organization for the Prohibition of Chemical Weapons listed ricinine as a schedule 1 toxic chemical [37].

Non-target screening over unexpected substances, neither target nor suspect compounds, identified a few hazardous chemicals in CPs and are shown in Figure 2. The spectra of all ions detected in the samples were compared with spectral databases, and four compounds that are unexpected in CPs were tentatively identified. Among the identified compounds, triethanolamine was detected in 28% of samples and was frequently found in laundry detergents (67%), shampoos (42%), and hair dyes (60%). It is used as a surfactant in cleaning formulation, such as soaps and detergents, and shows a relevant potential for human exposure. Due to low vapor pressure, dermal contact seems to be the most possible route of exposure, but inhalation is non-negligible in highly exposable situations (e.g., laundry room). Fortunately, triethanolamine shows low acute toxicity with LD50 values ranging from 4.19 to 11.26 g/kg, however, it is an irritant to the eyes and skin and causes systemic toxicity mainly in the kidneys, liver, red blood cells, and the nervous system following dermal and/or oral exposure in laboratory animals if not neutralized [38]. Iodocarb (IPBC) was detected in 6% of the samples and mostly found in body washes. Due to its good antimicrobial properties, it is frequently used as a preservative in CPs [39]. The maximum concentration of iodocarb allowed in cosmetics is 0.01%. A product exceeding the allowance level should be labeled “irritant” unless otherwise exempted. [40]. Galaxolidone was detected in 32% of the samples and frequently detected in face washes (41%), shampoos (67%), laundry detergents (40%), and fabric softeners (40%). It is indeed a metabolite of galaxolide used as fragrances or odorous components in a variety of CPs. It can concentrate in the blood, fat, and breast milk, and can affect androgen and progesterone receptors and also cause stimulation of estrogenic receptors in humans [41]. 2-(2H-Benzotriazol-2-yl)-4, 6-bis (1-methyl-1-phenylethyl) phenol was detected in 7% of the samples and was frequently found in laundry detergents. It is an ultraviolet (UV) stabilizer used in CPs for the prevention of UV light. Several countries have classified it as a Class I Specified Chemical Substance because of its potentially toxic and persistent nature and bioaccumulation properties [42].

Overall, diverse chemical additives, including biocides, UV filters, plasticizers, stabilizers, fragrances, and surfactants were found in CPs. These findings require human and environmental impact assessment. In Korea, a few studies have been conducted on the exposure of chemicals used in CPs, but was still insufficient to get a full insight of human risk, mainly due to the limited number of compounds used for the assessment. The suspect and non-target screening applied here provide still limited, but additional data, on the occurrence of chemical additives in CPs, which can help to extend knowledge on human exposures. Moreover, these extensive screenings are time and cost-effective, as no reference standards are required in advance. Considering that obtaining chemical standards and formulating stock solutions require preparative and practical time (e.g., a couple of months in general) the initiation and the duration for investigation with target screening would be hardly deterministic and thus often delayed more than one would expect [43]. The expected results would also be limited to target compounds whose reference standards are available. In the present study, we confirmed that suspect and non-target screening enable tentative identifications even for unexpected substances in CPs. It should be noted that novel chemical additives used in CPs are increasing either as newly created chemicals have been developed or as to replace substances with environmental/health issues that are restricted or prohibited by regulations. Under the circumstances, the conventional target screening definitively depends on the availabilities of reference standards and require preparation period for analytical setups (e.g., preparing stock solutions, plotting calibration curves, etc.) is insufficient to evaluate human exposure, in particular, when new additives are applied. On the other hand, suspect and non-target screening can provide rapid and confident results on the presence of suspected and unexpected substances without reference standards, as proved in the present study. These features make the suspect/non-target screening complementary for the target screening. It suggests that a combination of target and suspect/non-target screening is desirable to further understand the chemical exposure via consumption of CPs, resulting in a better assessment of human health risks. The common occurrence of these chemicals in various product categories potentially leads to higher exposure than expected. Thus, additional concerns on the newly identified chemical additives should be made in advance to avoid fatal health issues.

## 4. Conclusions

An integrated analytical procedure for target, suspect, and non-target screening based on liquid chromatography-high resolution mass spectrometry (LC-HRMS) with stepwise identification workflow was used for identification of known, suspect, and unexpected chemicals in consumer products. Among all analyzed CPs, isothiazolinones were found in 47% and phthalates in 24% of the samples. The concentration of target compounds in personal care products and cosmetics meet the Korean regulation while benzisothiazolone in washing agents, not being regulated, was found in relatively high concentrations (<518 mg/kg). For this reason, exposure and risk assessments for isothiazolinones should include washing agents that possibly remain in the laundry, leading to dermal exposure. Suspect and non-target analyses yielded six tentatively identified chemicals across the products including, benzophenone, ricinine, iodocarb (IPBC), galaxolidone, triethanolamine, and 2-(2H-Benzotriazol-2-yl)-4, and 6-bis (1-methyl-1-phenylethyl) phenol. Benzophenone, galaxolidone, and triethanolamine were successfully confirmed with reference standards, indicating that suspect and non-target screening can provide additional information on the suspected and unexpected substances in CPs without reference standards. The identification results revealed that selected CPs consistently contain diverse additives that are placed in a blind spot of regulative managements. Therefore, it is recommended that identification approaches, such as suspect and non-target screening should be undertaken to complement a conventional target analysis to widen a range of screening chemicals and to improve the quality of exposure and health risk assessments. 

## Figures and Tables

**Figure 1 ijerph-16-05075-f001:**
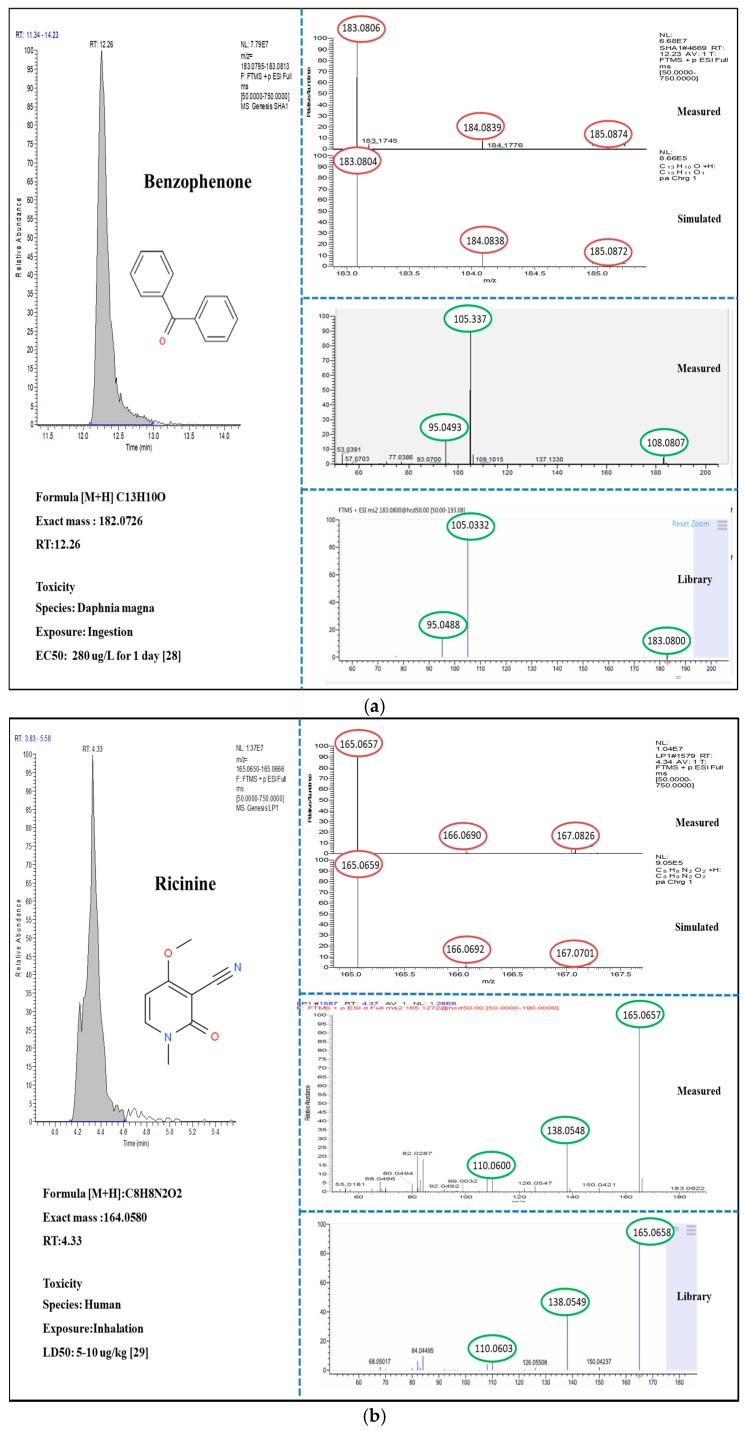
Chromatographic peak, isotopic pattern, and MS/MS spectra of benzophenone (**a**) and ricinine (**b**).

**Figure 2 ijerph-16-05075-f002:**
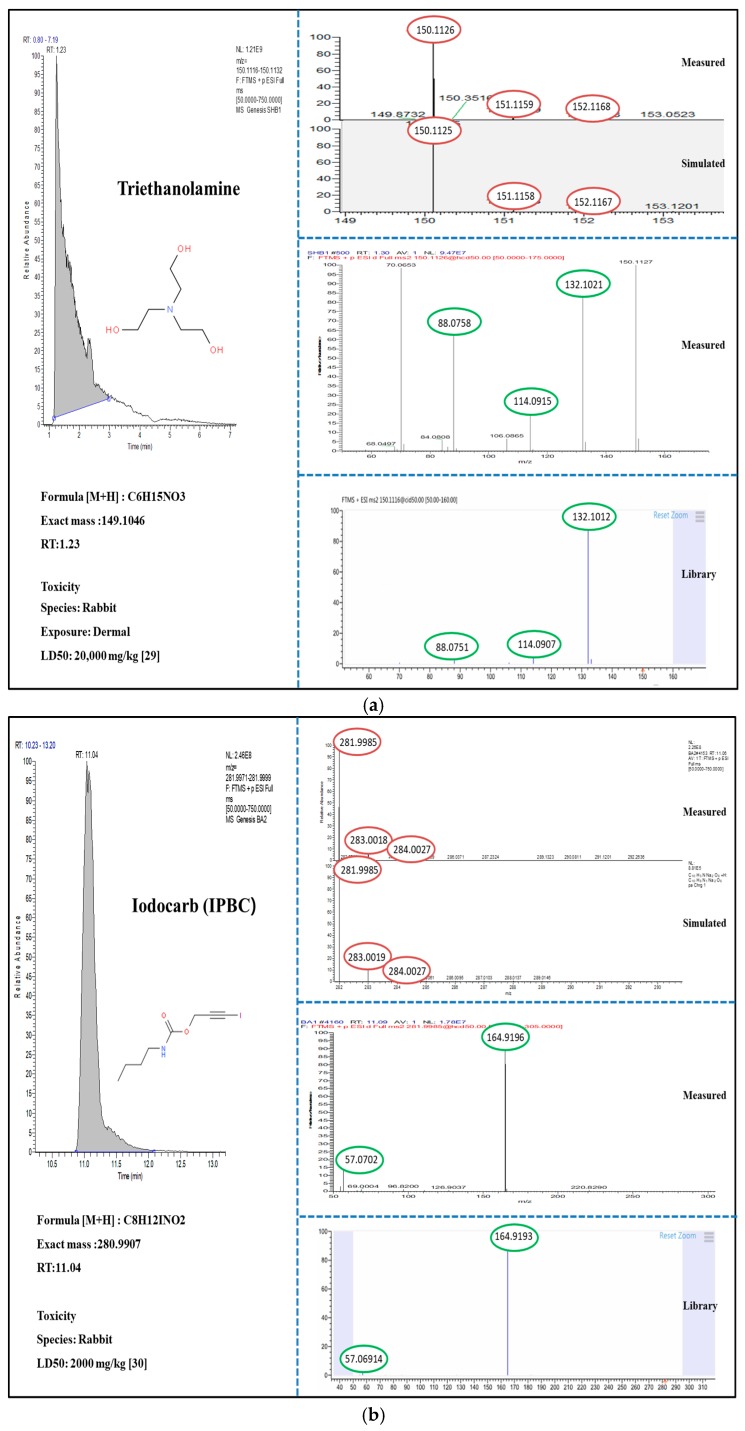
Chromatographic peak, isotopic pattern, and MS/MS spectra of triethanolamine (**a**), idocarb (IPBC) (**b**), galaxolidone (**c**), and 2-(2H-Benzotriazol-2-yl)-4,6-bis(1-methyl-1-phenylethyl) phenol (**d**), and their toxicity information if available elsewhere [28,29,30,31].

**Table 1 ijerph-16-05075-t001:** Concentrations and detection frequency (D.F) of the target compounds in chemical products (CPs) (mg/kg). MI: methylisothiazolinone, CMI: Methylchloroisothiazolinone, BIT: Benzisothiazolinone, DEP: Diethyl phthalate, and DMP: Dimethyl phthalate.

Products	MI	CMI	BIT	DEP	DMP
**Shampoos (*n* = 12)**					
SA1	- ^a^	-	-	<LOQ ^b^	<LOQ
SA2	-	-	-	-	0.26
SA3	-	-	-	0.4	-
SC1	-	-	0.165	-	-
D.F	0%	0%	8%	17%	17%
**Body Washer (*n* = 6)**					
BWA1	<LOQ	-	-	-	-
BWA2	<LOQ	-	-	-	-
BWA3	<LOQ	-	-	-	-
BWA4	<LOQ	-	-	-	-
BWA5	<LOQ	-	-	-	-
BWB1	<LOQ	-	-	-	-
D.F	100%	0%	0%	0%	0%
**Face Cleansers (*n* = 12)**					
FCA2	-	-	-	-	5.1
FCA3	-	-	-	-	7.9
FCA4	-	-	-	-	4.0
FCA5	-	-	-	-	4.1
FCC2	-	-	<LOQ	-	-
D.F	0%	0%	8%	0%	33%
**Lipstick (*n* = 5)**					
LP1	-	-	-	1.1	10
LP2	-	-	-	<LOQ	12
LP4	-	-	-	-	-
LP5	-	-	-	-	21
D.F	0%	0%	0%	40%	80%
**Hair Dyes (*n* = 5)**					
D.F	0%	0%	0%	0%	0%
**Dish Washer (*n* = 15)**					
DWB2	-	-	-	2.0	-
D.F	0%	0%	0%	7%	0%
**Laundry detergent (*n* = 15)**					
LDA1	-	-	103	-	-
LDA2	-	-	92	-	-
LDA3	-	-	103	-	-
LDA4	-	-	7.6	-	-
LDA5	-	-	49	-	-
LDB1	-	-	390	-	-
LDB2	-	-	509	-	-
LDB3	-	-	518	-	-
LDB4	-	-	419	-	-
LDB5	-	-	509	-	-
LDC1	-	-	125	-	-
LDC2	-	-	121	-	-
LDC3	-	-	78	-	-
LDC4	-	-	68	-	-
LDC5	-	-	1.2	-	-
D.F	0%	0%	100%	0%	0%
**Fabric Softener (*n* = 15)**					
FSA1	1.1	1.47	49	1.1	-
FSA2	-	1.22	41	-	<LOQ
FSA3	-	1.37	41	-	<LOQ
FSA4	-	-	38	-	<LOQ
FSA5	-	-	39	-	<LOQ
FSB1	-	<LOQ	46	-	-
FSB2	-	<LOQ	44	-	-
FSB3	-	<LOQ	43	-	-
FSB4	-	<LOQ	48	-	-
FSB5	-	8.0	0.62	-	-
D.F	53%	0%	67%	7%	27%
**All CPs**					
Median	1.4	-	49	1.0	5.1
Average	3	-	134	1.1	7.3
Maximum	7.1	-	518	2.0	21.2
Minimum	1.2	-	0.2	0.4	0.26
D.F	15%	0%	32%	7%	16%

a: Not Detected; b: Below limit of quantification.

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
