# Peer review of "Occurrence and Concentration of Chemical Additives in Consumer Products in Korea"

_ijerph, 2019, doi:10.3390/ijerph16245075_

Round 1

Reviewer 1 Report

The presented research is very interesting. The authors should also note who would be interested in performing such tests to the CPs. Maybe some governmental chemical laboratory that would check the products?

Minor English spell check is required in some fields:

line 25: in 47% and in 24% of the samples

line 30: selected CPs consistently contain

lines 36-37: please rephrase

line 186: "3.1. Quality assurance and quality control"

Why do you mention in section 3 the outcome of the QA/QC? In my opinion this should be analysed in the Methodology section at the respective paragraph(2.5).

line 241: the relatively high concentrations of phthalates found in lipsticks and face cleanser are probably...

lines 275-276: "As the result....above." Please rephrase.

lines 280-282: "Thus, there.....leading to humans ingestion". Please rephrase.

lines 286-287: close the parenthesis 

line 312: was detected in 6% of the samples

lines 322-323: Several countries have classified it ...

line 402-403: in 24% of the samples

line 413: selected CPs consistently contain...

Author Response

Comments/Suggestion

Responses

Minor English spell check is required in some fields:

Line 25: in 47% and in 24% of the samples

Line 30: selected CPs consistently contain

Corrected according to the comments within the revised manuscript as below

Line 26, “in 47 % and in 24 % of the samples”.

Line 31, “selected CPs consistently contains”

Lines 36-37: please rephrase

It is now rephrased as below.

Line 37-40, “The development of chemical industry from in the last few decades past century has supplied the world an extensive amount introduced the world with a vast amount of chemicals. Recently Presently there are approximately 100,000 chemicals being used globally and over 500 new chemicals are introduced produced annually”

Line 186: "3.1. Quality assurance and quality control"

Why do you mention in section 3 the outcome of the QA/QC? In my opinion this should be analyzed in the Methodology section at the respective paragraph (2.5).

The outcome of QA and QC were moved to methodology section at line 136-147.

Line 241: the relatively high concentrations of phthalates found in lipsticks and face cleanser are probably...

Corrected at line 251-252, “The relatively high concentrations of phthalates found in lipsticks and face cleanser found is are probably”.

Lines 280-282: "Thus, there......leading to humans ingestion". Please rephrase.

Corrected at line 290-291, “Thus, there has been concerned about the leaching of benzophenone leached from the packaging material”

Lines 286-287: close the parenthesis

Corrected at line 297

Line 312: was detected in 6% of the samples

Corrected at line 323, “detected in 6 % of the samples”

Lines 322-323: Several countries have classified it ...

Corrected at line 334, “Several countries have been classified”

Line 402-403: in 24% of the samples

Corrected at line 413, “in 24% of the samples”

Line 413: selected CPs consistently contain...

Corrected at line 424, “selected CPs consistently contain”

Reviewer 2 Report

In the paper  authors presented the results of detection and determination of chemical additives in CP. For identification of known, suspect and unknown chemicals in consumer products including cosmetics, personal care products, and washing agents they applied an integrated target/suspect/non-target screening procedure using liquid chromatography-high resolution mass spectrometry (LC-HRMS). The obtained results are interesting.

Report is good write, however, some improvement can be applied.  I suggest only minor revision.

The references in the text have no consecutive numbers:

line 51 [20]

151 [36] 207 [42] 223 [22] 225 [21] 243 [43] 265 [44] 309 [45]

112 –How samples were shaken? Please add adequate information 123-125- Please indicate the share of which solvent increased during the gradient elution. Table 1 – In my opinion, it makes no sense to give of median, average, maximum and minimum values, if a given compound was detected/ determined in only one product. Maybe in the line with the name of the product and its quantity should be added the number of products in which the compound was determined. Table 1. – the value of DMP in lipstick are not correct as average is higher than maximum. The order of compounds in Table S2 and S3 in Supplementary Materials should be the same as in Table 1. The reference 38-41 were not mentioned in the text Please, prepare “References” according to the guidelines of the journal. There are a lot of mistakes.

[5] The lack of no of volume and pages

[5,6, 12,16]- the name of journal as a title

[13,14,15,20,24] – the journal name is missing

[7, 23] correct the authors' name

[17] should be “Identification …

Author Response

Comments/Suggestion

Responses

The references in the text have no consecutive numbers:

line 51 [20]

151 [36] 207 [42] 223 [22] 225 [21] 243 [43] 265 [44] 309 [45]

All references were revised as pointed out.

112 –How samples were shaken? Please add adequate information 123-125- Please indicate the share of which solvent increased during the gradient elution.

Information were added to the manuscript at line 114 (“in an orbital shaker”) and line 127-130 (“The gradient elution started at 5% of (B) increased to 75% at 10 min then the content of B component was further increased to 95% at 15 min and this condition was held for 5 min, following this mobile phase composition was set-back to initial conditions and maintained for 10 min to equilibrate the column.”)

Table 1 – In my opinion, it makes no sense to give of median, average, maximum and minimum values, if a given compound was detected/ determined in only one product. Maybe in the line with the name of the product and its quantity should be added the number of products in which the compound was determined.

Table 1 is changed according to the comments

Table 1. – the value of DMP in lipstick are not correct as average is higher than maximum.

The value is corrected

The order of compounds in Table S2 and S3 in Supplementary Materials should be the same as in Table 1.

The order of the compounds in Table S2 and S3 were changed and now are same as Table 1

The reference 38-41 were not mentioned in the text

All the references were revised, references from 40-43 are not mentioned in the text but these are the references used for the toxicity values in the figures.